# Catalytic Combustion of Dimethyl Disulfide on Bimetallic Supported Catalysts Prepared by the Wet-Impregnation Method

**Junan Gao [1], Song Gao [1], Jun Wei [2], Hong Zhao [1] and Jie Zhang [1,\*]**

[1] State Key Laboratory of Chemical Resource Engineering, Beijing University of Chemical Technology, Beijing 100029, China; gaojunan321@163.com (J.G.); gaosong@mail.buct.edu.cn (S.G.); zhaohong@mail.buct.edu.cn (H.Z.)

[2] Hangzhou Yinli Environmental Protection Technology Co., Ltd., Hangzhou 310016, China; weijun123@163.com

[\*] Correspondence: zhangjie@mail.buct.edu.cn; Tel.: +86-180-0113-7991

 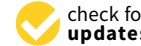

**Abstract:** In this paper, the catalytic combustion of DMDS (dimethyl disulfide, $CH_3SSCH_3$) over bimetallic supported catalysts were investigated. It was confirmed that $Cu/\gamma\text{-}Al_2O_3\text{-}CeO_2$ showed best catalytic performance among the five single-metal catalysts. Furthermore, six different metals were separately added into $Cu/\gamma\text{-}Al_2O_3\text{-}CeO_2$ to investigate the promoting effect. The experiments revealed Pt as the most effective promoter and the best catalytic performance was achieved as the adding amount of 0.3 wt%. The characterization results indicated that high activity and resistance to sulfur poisoning of $Cu\text{-}Pt/\gamma\text{-}Al_2O_3\text{-}CeO_2$ could be attributed to the synergistic effect between Cu and Pt.

**Keywords:** catalytic combustion; dimethyl disulfide; bimetallic; supported catalyst

## 1. Introduction

Sulfur containing volatile organic compounds (SVOCs) emissions are exhaust gases containing organic sulfur compounds, such as thiophene, mercaptans, and thioethers. Almost all hydrocarbons contain organic sulfur compounds in their raw materials, a large number of SVOCs will be produced in the process of dye manufacturing, pesticide production, coatings industry, leather production, landfill and wastewater treatment [1–3]. Some exhaust gases, containing organic sulfur compounds, can cause malodors and damage to skin and eyes, even at low concentrations [4–7]. At present, there are many SVOCs treatment methods: condensation method, adsorption method, catalytic combustion method, low temperature plasma purification method, biological method et al. [8–12]. Catalytic combustion SVOCs will produce $CO_2$, $H_2O$, $SO_2$ and other compounds, supplemented by an exhaust gas absorption device, which can completely eliminate SVOCs. It is a promising SVOCs treatment method. Because DMDS (dimethyl disulfide, $CH_3SSCH_3$) is among the most odorous compounds due to its low human detection threshold (2.5 μg/m$^3$), which makes it rather difficult to treat completely [13–15]. DMDS was selected reactant of this study.

There are many supports for catalytic combustion of DMDS: $\gamma\text{-}Al_2O_3$, $CeO_2$, $SiO_2$, $TiO_2$, $ZrO_2$, ZSM-5 [16–18]. These supports, with a high specific surface area, increase the dispersibility of the metal and the adsorption capacity of the reactants and reduce the loading of the metal [19,20]. $\gamma\text{-}Al_2O_3$ and $CeO_2$ are the most widely studied and applied due to their high specific surface area, stability, and low price [21,22]. Therefore, this study used these two supports, $CeO_2$ and $\gamma\text{-}Al_2O_3$, in experiments. The catalysts that are commonly used for DMDS catalytic combustion to improve catalytic activity are supported noble metals, transition metal oxides, and bimetallic oxides catalysts [23–25]. Noble

metal supported catalysts have long been considered to be desirable for catalytic combustion of VOCs due to their higher specific activity and regenerability [23,26]. However, the use of Pt- and Pd-based catalysts is limited in applications dealing with organic chlorine and organic sulfur exhaust gas, while few studies using Au in DMDS or chlorinated VOC oxidation have shown evidence of sulfur and chlorine resistance [27]. More researchers are working on catalytic combustion of DMDS in transition metal catalysts, replacing noble metal catalysts [28]. Oxides of transition metals, mainly Mn, Co, Fe, and Cu, are employed for the combustion of DMDS, and Wang et al. researched effect of acid treatment on the performance of the $CuO–MoO_3/Al_2O_3$ catalyst for the destructive oxidation of $(CH_3)_2S_2$ [29].

At present, many monometal and bimetallic catalyst studies by the group of Wang, Nevanperä, and Keiski, from as early as the 1990s, have reported highly active and $SO_2$ selective catalysts in DMDS oxidation, such as $Cu/CeO_2–Al_2O_3$, $Pt/Al_2O_3–CeO_2$, and $Pt–Cu/Al_2O_3–SiO_2$. Keiski et al. found doping of the $Al_2O_3$ substrate by $SiO_2$ led to a more selective and stable $Pt–Cu/Al_2O_3$ catalyst and the stability test by Nevanperä et al. for $Au/CeO_2–Al_2O_3$ showed that the catalyst is durable for over 40 h [10,27,30]. In the process of catalytic combustion of DMDS, sulfur compounds bond tightly to the active site of the catalyst, forming stable surface metal sulfides which prevent adsorption of reactants on the surface. Sulfur can also concentrate on the oxide support, forming sulfates such as aluminum sulfates and oxysulfates, which can have an impact on the metal–support interaction [31]. Nevanperä et al. described an investigation related to preparation of Pt–Au and Cu–Au bimetallic catalysts, which showed good performance in catalytic combustion of DMDS [23]. Therefore, the metal synergy effect between bimetallic catalysts deserves further research.

In this study, the metal oxide and support of catalyst was screened based on the works in the literature mentioned above. The most active of the numerous $γ-Al_2O_3–CeO_2$ supported single-metal oxides was determined as the principal catalyst. A variety of metal oxides were separately added to this catalyst to investigate the promoting effect, and the most effective one was selected as a promoter in later investigation. The physicochemical properties of the catalyst, structure–activity relationships, and reaction mechanism were studied using different characterization techniques. $Cu–Pt/γ-Al_2O_3–CeO_2$ was screened as the best catalyst for the catalytic combustion of DMDS. The effects of preparation conditions on the catalytic performance of $Cu–Pt/γ-Al_2O_3–CeO_2$ were investigated, and 1000 h stability experiments were performed.

## 2. Results and Discussion

### 2.1. Effect of the Supports

In order to study the effect of different supports on the activity of supported catalysts, $Cu/(γ-Al_2O_3, CeO_2, γ-Al_2O_3-CeO_2)$ and $Cu-Pt/(γ-Al_2O_3, CeO_2, γ-Al_2O_3-CeO_2)$ catalysts were prepared, the performance of catalytic combustion DMDS were investigated, and the results are shown in Figure 1. Under the experimental conditions, the DMDS catalytic ignition temperatures of $Cu/γ-Al_2O_3-CeO_2$, $Cu/γ-Al_2O_3$ and $Cu/CeO_2$ catalysts are 200 °C–300 °C. The catalytic activity of DMDS is compared between three Cu-based catalysts and the order of activity: $Cu/γ-Al_2O_3-CeO_2 > Cu/γ-Al_2O_3 > Cu/CeO_2$. Moreover, it can be seen that the activity of Pt-Cu bimetallic supported catalyst is consistent with that of Cu single-metal supported catalyst. The catalytic activity of $Cu-Pt/γ-Al_2O_3-CeO_2$ catalyst is significantly higher than that of $Cu-Pt/γ-Al_2O_3$ and $Cu-Pt/CeO_2$. Compared with $γ-Al_2O_3$, $γ-Al_2O_3-CeO_2$ has great advantages in catalytic combustion of DMDS for the addition of $CeO_2$. Because $CeO_2$ has a very strong oxygen storage capacity, and the transfer of charge between the active species and the $CeO_2$ support is beneficial to enhance the reactivity of the catalyst [32]. Therefore, $γ-Al_2O_3-CeO_2$ was chosen as the catalyst support in this study.

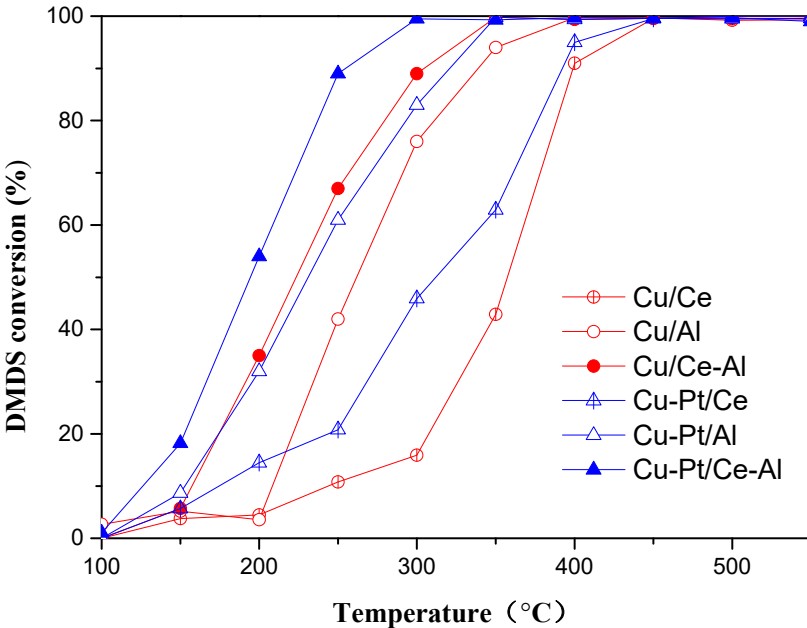

**Figure 1.** Activity diagram of catalytic combustion of DMDS by Cu/($\gamma$-Al$_2$O$_3$, CeO$_2$, $\gamma$-Al$_2$O$_3$-CeO$_2$) and Cu-Pt/($\gamma$-Al$_2$O$_3$, CeO$_2$, $\gamma$-Al$_2$O$_3$-CeO$_2$) catalysts. Al is $\gamma$-Al$_2$O$_3$, Ce is CeO$_2$, and Ce-Al is $\gamma$-Al$_2$O$_3$-CeO$_2$, which is the same as the following figure.

The X-ray diffractometer (XRD) patterns of Cu/($\gamma$-Al$_2$O$_3$, CeO$_2$, $\gamma$-Al$_2$O$_3$-CeO$_2$) and Cu-Pt/($\gamma$-Al$_2$O$_3$, CeO$_2$, $\gamma$-Al$_2$O$_3$-CeO$_2$) catalysts are showed in Figure 2, in which we can observe the dispersibility of the active component on the support from a microscopic point of view. The diffraction peak of CuO on the Cu/CeO$_2$ catalyst can be clearly observed, indicating that the Cu phase on the surface of the CeO$_2$ support is in the form of larger CuO, which is more unfavorable for the catalytic combustion reaction. Therefore, CeO$_2$ supported single-metal catalysts have the worst activity. With no diffraction peaks of CuO appearing in the Cu-Pt/CeO$_2$ catalyst, it can be inferred that the addition of Pt causes CuO to form smaller particle grains and improve the distribution of CuO. In addition, with comparing the Cu/$\gamma$-Al$_2$O$_3$-CeO$_2$ and Cu/$\gamma$-Al$_2$O$_3$ XRD patterns, it was found that the addition of CeO$_2$ enhanced the regularity of the Al$_2$O$_3$ micropores and facilitated the dispersion of CuO. At the same time, it can be clearly seen that the composite support of $\gamma$-Al$_2$O$_3$-CeO$_2$ has a higher crystallinity than the CeO$_2$ and Al$_2$O$_3$ single support. CeO$_2$ promotes the migration of lattice oxygen and improves the catalytic combustion efficiency, so the catalytic activity of the composite support is higher than that of the single-support catalyst, a bimetal or a single metal. The XRD patterns of Cu-Pt/($\gamma$-Al$_2$O$_3$, CeO$_2$, $\gamma$-Al$_2$O$_3$-CeO$_2$) do not show the diffraction peaks of Cu phase or Pt phase. It can be inferred that Cu phase or Pt phase in support are present in the form of oxides of highly dispersed small particles, which is advantageous for the reaction [33].

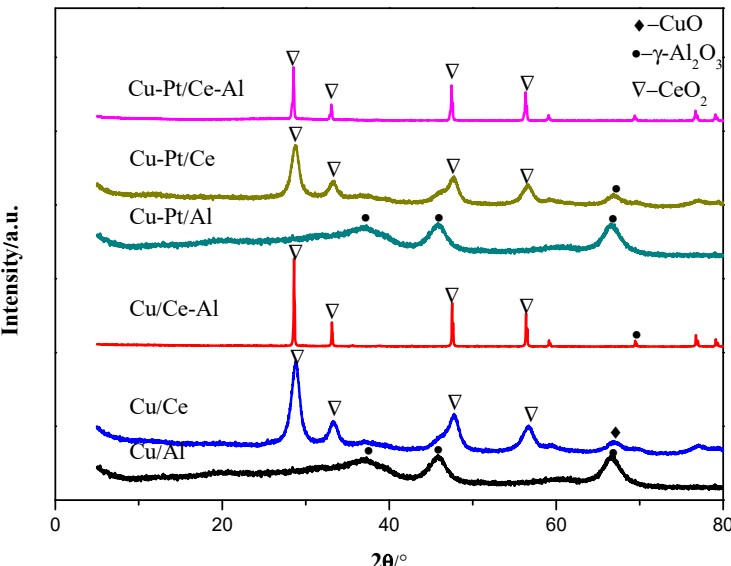

**Figure 2.** X-ray diffractometer (XRD) patterns of Cu/($\gamma$-Al$_2$O$_3$, CeO$_2$, $\gamma$-Al$_2$O$_3$-CeO$_2$) and Cu-Pt/ ($\gamma$-Al$_2$O$_3$, CeO$_2$, $\gamma$-Al$_2$O$_3$-CeO$_2$) catalysts.

### 2.2. Single-Metal Supported Catalyst

In order to select the principal catalyst, activity of single-metal supported catalyst was evaluated, and the characteristic temperature diagram of catalytic combustion of DMDS of (Cu, Fe, Zn, Mo, V)/$\gamma$-Al$_2$O$_3$-CeO$_2$ catalyst, are showed in Figure 3. The complete conversion temperature of Cu/$\gamma$-Al$_2$O$_3$-CeO$_2$ is about 308 °C, which is obviously superior to other catalysts. The complete conversion temperature of V/$\gamma$-Al$_2$O$_3$-CeO$_2$ and Zn/$\gamma$-Al$_2$O$_3$-CeO$_2$ catalysts is about 345 °C, and the catalytic effect is not obvious. At the same time, T$_{50}$ (temperature at which DMDS conversion rate is 50%) is the main evaluation condition, supplemented by T$_{10}$ (temperature when DMDS conversion rate is 10%) and T$_{100}$ (temperature when DMDS conversion rate is 100%). The DMDS catalytic activity of a transition metal catalyst is obtained in order of activity from high to low: (Cu > Fe > Mo > V > Zn)/$\gamma$-Al$_2$O$_3$-CeO$_2$. Based on the above analysis, the catalytic activity of Cu/$\gamma$-Al$_2$O$_3$-CeO$_2$ catalyst for oxidizing DMDS is the highest in the transition metal $\gamma$-Al$_2$O$_3$-CeO$_2$ catalyst, and was served as the principal catalyst for further investigation.

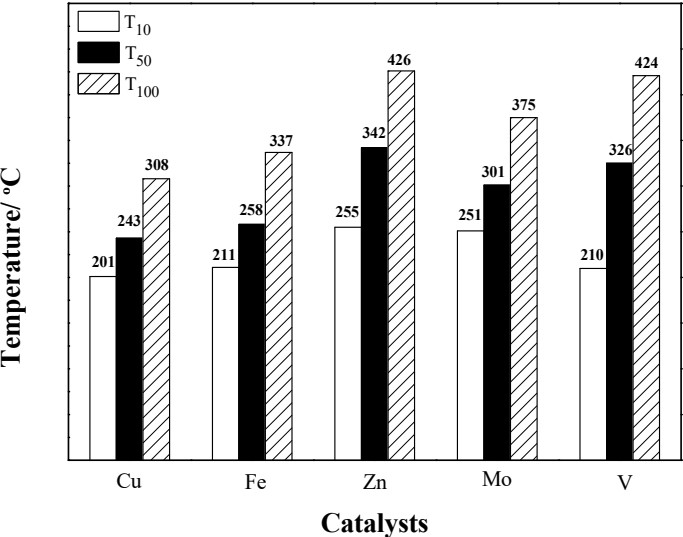

**Figure 3.** Characteristic temperature diagram of catalytic combustion of DMDS with (Cu, Fe, Zn, Mo, V)/$\gamma$-Al$_2$O$_3$-CeO$_2$ catalyst.

Figure 4 shows the XRD patterns of (Cu, Fe, Zn, Mo, V)/$\gamma$-Al$_2$O$_3$-CeO$_2$ catalyst, in which we can know the effect of the dispersibility of the active component on the catalytic activity. It can be found that after metal supported, such as Cu, Fe, Zn, Mo, and V, the diffraction peak of the support patterns are clearly visible, and the position of the peak does not change, indicating preparation did not modify the ordered structure typical of $\gamma$-Al$_2$O$_3$-CeO$_2$ [34,35]. Only ZnO is present in the form of large particles on the support, which diffraction peaks are clearly visible, so its catalytic combustion activity is lowest. The intensity of diffraction peaks of (Fe, Zn, Mo, V)/$\gamma$-Al$_2$O$_3$-CeO$_2$ catalysts is slightly decreased, and the intensity of Cu/$\gamma$-Al$_2$O$_3$-CeO$_2$ diffraction peaks is slightly increased, indicating that the catalyst of Cu/$\gamma$-Al$_2$O$_3$-CeO$_2$ has fewer lattice defects and enhances the order of its support [36].

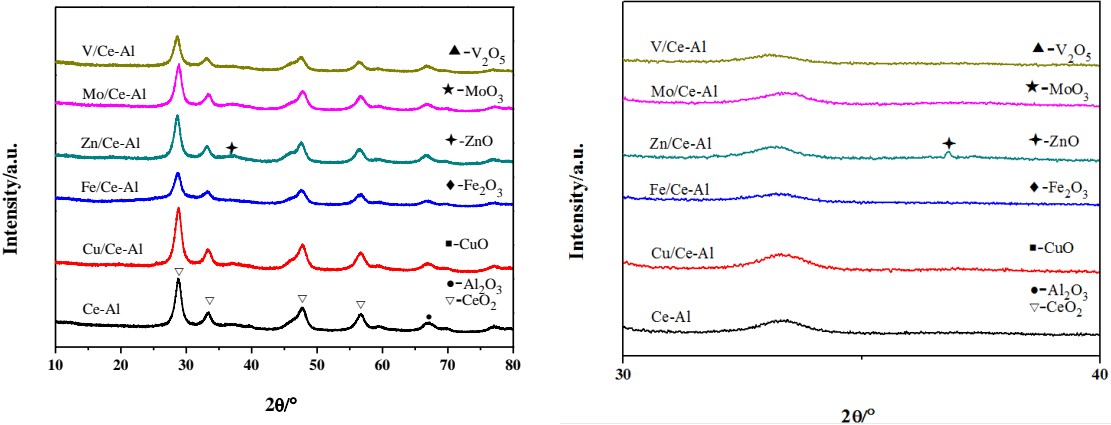

**Figure 4.** X-ray diffractometer (XRD) patterns of (Cu, Fe, Zn, Mo, V)/$\gamma$-Al$_2$O$_3$-CeO$_2$ catalyst. On the right is the enlarged view on the left.

The temperature programmed reduction (TPR) patterns can effectively show the hydrogen consumption of the supported oxide reduction, the ease of reduction, and the interaction between the metal oxide and the support. The H$_2$-TPR patterns of the (Cu, Fe, Zn, Mo, V)/$\gamma$-Al$_2$O$_3$-CeO$_2$ catalyst are showed in Figure 5. $\gamma$-Al$_2$O$_3$-CeO$_2$ catalyst showed a reduction peak at 294 °C, which can be attributed to CuO reduction [37]. By comparison, it was found that the reduction peak of Cu/$\gamma$-Al$_2$O$_3$-CeO$_2$ catalyst had the lowest temperature, the highest peak intensity, and the best reduction. The redox capacities of the catalysts are ranked: (Cu > Fe > Mo > V > Zn)/$\gamma$-Al$_2$O$_3$-CeO$_2$, which is highly consistent with the catalytic combustion activity of the transition metal catalyst DMDS. It is inferred that the stronger the redox ability of the catalyst, the stronger the catalytic combustion activity of DMDS.

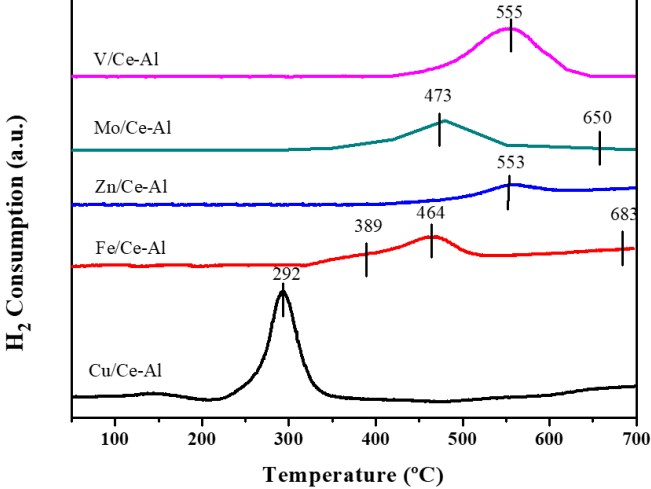

**Figure 5.** H$_2$-TPR patterns of (Cu, Fe, Zn, Mo, V)/$\gamma$-Al$_2$O$_3$-CeO$_2$ catalyst.

### 2.3. Bimetallic Catalyst

Cu/$\gamma$-Al$_2$O$_3$-CeO$_2$ catalyst was determined as the principal catalyst, due to its low redox temperature and the best activity of catalytic combustion of DMDS in transition metals. A variety of metal oxides (Mo, Fe, Zn, V, Pt, Pd) were separately added to this catalyst to investigate the promoting effect. Figure 6 shows the characteristic temperature of catalytic combustion of DMDS by (Cu, Cu-Mo, Cu-Fe, Cu-Zn, Cu-V, Cu-Pt, Cu-Pd)/$\gamma$-Al$_2$O$_3$-CeO$_2$ catalysts. Compared with the catalytic activity of Cu/$\gamma$-Al$_2$O$_3$-CeO$_2$, when the promoter is a transition metal, it is found that Cu-Mo/$\gamma$-Al$_2$O$_3$-CeO$_2$ is superior to other catalysts, and T$_{100}$ is about 285 °C. The addition of Mo metal increased the catalytic activity of Cu/$\gamma$-Al$_2$O$_3$-CeO$_2$. After the addition of Zn and Fe, the catalytic activity of DMDS was not significantly improved, while the catalytic activity of Cu-V/$\gamma$-Al$_2$O$_3$-CeO$_2$ catalyst was 340 °C and T$_{100}$ was about 449 °C. It can be seen that the addition of V inhibits the catalytic activity of Cu/$\gamma$-Al$_2$O$_3$-CeO$_2$. When promoter was a noble metal, it was found that (Pt, Pd)/$\gamma$-Al$_2$O$_3$-CeO$_2$ had good activity for catalytic combustion of DMDS and Pt-Cu/$\gamma$-Al$_2$O$_3$-CeO$_2$ was more significant.

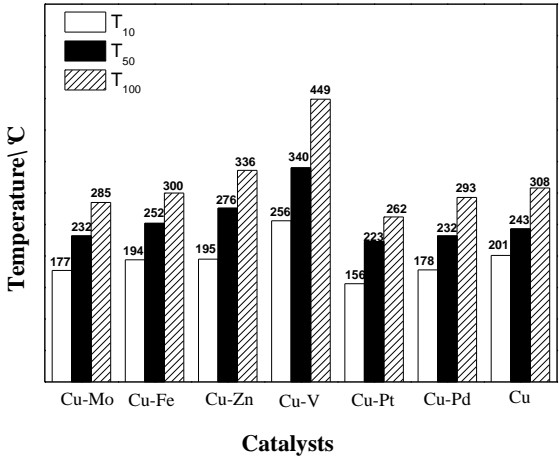

**Figure 6.** Catalytic combustion temperature of DMDS of (Cu, Cu-Mo, Cu-Fe, Cu-Zn, Cu-V, Cu-Pt, Cu-Pd)/$\gamma$-Al$_2$O$_3$-CeO$_2$ catalysts.

In order to evaluate the ability of catalyst to resist sulfur poisoning, the stability of DMDS for 48 h (Cu, Pt, Cu-Mo, Cu-Fe, Cu-Zn, Cu-V, Cu-Pt, Cu-Pd)/$\gamma$-Al$_2$O$_3$-CeO$_2$ catalysts are showed in Figure 7. The evaluation conditions were: GHSV (gaseous hourly space velocity) of 50,000 h$^{-1}$, reaction temperature of 300 °C, and DMDS concentration of 1000 ppm. The results show, the addition of (Mo, Pt, Pd) metal enhances the sulfur resistance of Cu/$\gamma$-Al$_2$O$_3$-CeO$_2$. But, only Cu-Pt/$\gamma$-Al$_2$O$_3$-CeO$_2$ catalytic combustion DMDS activity is always maintained at about 100%, catalytic activity of Pt/$\gamma$-Al$_2$O$_3$-CeO$_2$ is rapidly inactivated for poor resistance to sulfur poisoning. The reason for catalyst deactivation is that sulfur compounds bond tightly to the active site of the catalyst forming stable surface metal sulfides during catalytic combustion, which prevent adsorption of reactants on the surface [38–40]. It can be seen from Figure 10 that the Pt phsae in the Cu-Pt/CeO$_2$-Al$_2$O$_3$ catalyst exists in the form of Pt$^0$. The presence of Pt$^0$ can enhance the dispersibility, reduction and sulfur poisoning ability of CuO than PtO [41]. Combined with Figure 6, it is found that the catalytic activity of Cu-Pt/$\gamma$-Al$_2$O$_3$-CeO$_2$ catalyst DMDS is the highest, and the good sulfur poisoning ability. Cu-Pt/$\gamma$-Al$_2$O$_3$-CeO$_2$ catalyst is an ideal catalyst for catalytic combustion of DMDS.

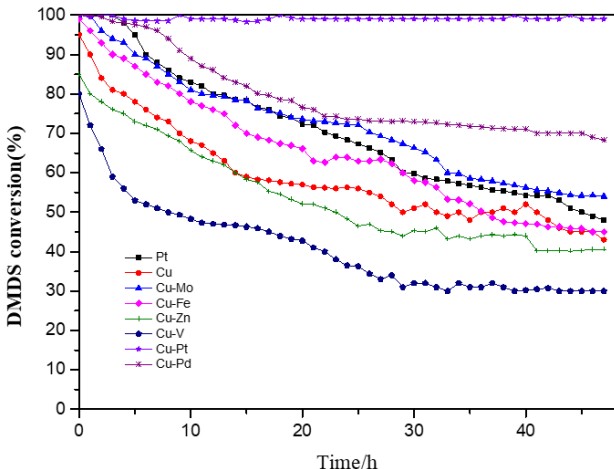

**Figure 7.** (Cu, Cu-Mo, Cu-Fe, Cu-Zn, Cu-V, Cu-Pt, Cu-Pd)/CeO$_2$-Al$_2$O$_3$ catalysts for catalytic combustion of DMDS 48 h stability diagram.

Figure 8 shows the XRD patterns of (Cu, Cu-Mo, Cu-Fe, Cu-Zn, Cu-V, Cu-Pt, Cu-Pd)/γ-Al$_2$O$_3$-CeO$_2$ catalysts, in which we can obtain information of the structure or morphology of the molecules inside the material. Cu, Cu-Mo, Cu-Fe, Cu-Zn, Cu-V Cu-Pt, Cu-Pd supported on the γ-Al$_2$O$_3$-CeO$_2$ support, the diffraction peak of the γ-Al$_2$O$_3$-CeO$_2$ support pattern is clearly visible, and the position of the peak does not change [42]. It indicates that the support retains its structure intact after metal ion impregnation. However, after the addition of Fe, Zn, Mo, V, Pt, and Pd, the intensity of the diffraction peak of the bimetallic catalyst is lower than that of the Cu/γ-Al$_2$O$_3$-CeO$_2$ catalyst. It indicates that the addition of Fe, Zn, Mo, V, Pt, and Pd changes the order of Cu/γ-Al$_2$O$_3$-CeO$_2$ and reduces the crystallinity of γ-Al$_2$O$_3$-CeO$_2$. Only the characteristic diffraction peaks of Cu-Zn/γ-Al$_2$O$_3$-CeO$_2$ catalysts are observed, which catalytic activity and sulfur poisoning ability are not good, other metal oxide characteristic peaks are not found. This may be because the impregnation process and the calcination process are good for the catalyst treatment and the Cu, Fe, Mo, V, Pt, and Pd phase have small particle size and are highly dispersed [43].

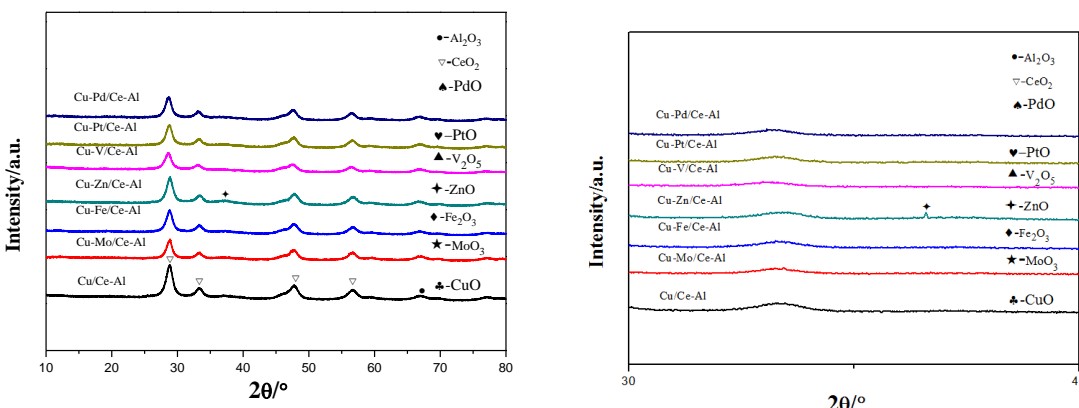

**Figure 8.** XRD patterns of (Cu, Cu-Mo, Cu-Fe, Cu-Zn, Cu-V, Cu-Pt, Cu-Pd)/γ-Al$_2$O$_3$-CeO$_2$ catalysts. On the right is the enlarged view on the left.

In order to obtain the information on the interaction between metal oxides, or metal oxides and supports, during the reduction process of supported metal catalysts, H$_2$-TPR patterns of (Cu, Cu-Mo, Cu-Fe, Cu-Zn, Cu-V, Cu-Pt, Cu-Pd)/γ-Al$_2$O$_3$-CeO$_2$ catalysts are showed in Figure 9. It is found that the reduction peak of Cu-Pd/γ-Al$_2$O$_3$-CeO$_2$ catalyst at 166 °C is attributed to the highly dispersed PdO, and there is a significant reduction peak at 271 °C, which belongs to the reduction of CuO and the small particles of CeO$_2$ with highly dispersed catalyst surface reduction [44]. The loading of Pt reduced

the temperature of Cu/γ-Al₂O₃-CeO₂ support reduction, enhanced the strength of the reduction peak, and enhanced its reducibility. This indicates that the effect of Pt on the surface of Cu-Pt/γ-Al₂O₃-CeO₂ on the reduction of CeO₂ is affected by CeO₂ hydrogen spillover effect [45]. The loading of Mo, Fe and Zn metals enhances the strength of the reduction peak of Cu-M/γ-Al₂O₃-CeO₂ catalyst, and the loading of V greatly reduces the reducibility of Cu-V/γ-Al₂O₃-CeO₂ catalyst.

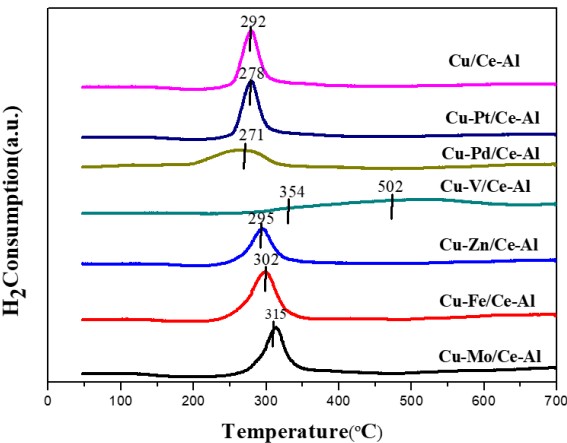

**Figure 9.** H₂-TPR patterns of (Cu, Cu-Pd, Cu-Pt, Cu-Mo, Cu-Fe, Cu-Zn, Cu-V)/γ-Al₂O₃-CeO₂ catalyst.

In order to obtain the chemical and electronic states of the medium metal of the bimetallic catalyst, Cu2p of (Cu-Pd, Cu-Pt, Cu-Mo, Cu-Fe, Cu-Zn, Cu-V, Cu)/γ-Al₂O₃-CeO₂ catalyst are showed in Figure 10a. By comparison, it can be found that all the catalysts have three peaks, the main peak at BE = 934.5 eV, corresponding to the characteristic peak of Cu2p₂/₃ orbital. The satellite peaks at BE = 943.5 eV and BE = 954.1 eV belong to the characteristic peak of Cu2p₁/₂, and there is no peak shift. The above information indicates that Cu phase on the surface of (Cu-Pd, Cu-Pt, Cu-Mo, Cu-Fe, Cu-Zn, Cu-V, Cu)/γ-Al₂O₃-CeO₂ catalysts mainly exist in the form of CuO [46]. Figure 10b shows XPS spectrum of a Cu-Pt/γ-Al₂O₃-CeO₂ catalyst. It can be found that the peaks of Cu-Pt/γ-Al₂O₃-CeO₂ catalyst Pt4f at BE = 71.2 eV and BE = 74.65 eV are Pt4f₇/₂ and Pt4f₅/₂, respectively, which can be attributed to Pt⁰ [47]. The shoulder peak at BE = 72.8 eV can be attributed to PtO on the surface of the catalyst.

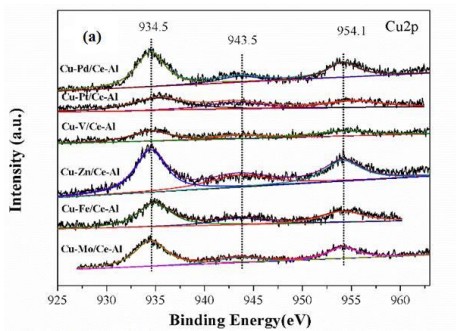 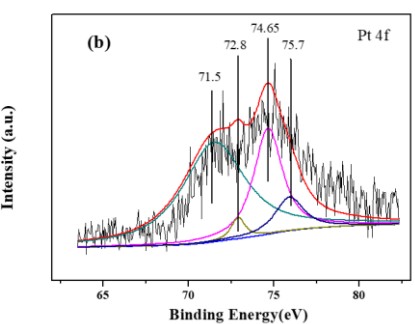

**Figure 10.** (**a**) Catalyst Cu2p of (Cu-Pd, Cu-Pt, Cu-Mo, Cu-Fe, Cu-Zn, Cu-V, Cu)/γ-Al₂O₃-CeO₂, (**b**) Pt4f XPS spectrum of Cu-Pt/γ-Al₂O₃-CeO₂.

The right amount of loading is critical to the impact of the supported catalyst, so Figure 11 shows the catalytic combustion activity of DMDS for different loadings of Cu-Pt/γ-Al₂O₃-CeO₂ catalyst. It can be seen from the Figure 11a that as the Cu loading increases from 0% to 10%, the DMDS catalytic combustion activity of the Cu catalyst gradually increases. As the Cu loading increases, the trend of catalyst activity growth decreases. The possible reasons are as follows: the specificity of the specific surface area of the catalyst carrier itself is fixed. At the beginning of loading of Cu increased,

more active sites can be introduced, which is beneficial to increasing the catalytic combustion activity of DMDS. However, when the dispersibility of Cu species reaches the threshold, increasing the loading of Cu introduce more active sites, and the excessive Cu loading cause the clustering effect of the metal. And the catalytic combustion activity of DMDS, it is impossible to be further improved. It can be seen from Figure 11b that as the Pt loading increases from 0% to 0.5%, the DMDS catalytic combustion activity of the catalyst gradually increases. However, as the Pt loading was from 0.3% to 0.5%, there was almost no change in catalyst activity growth. In summary, when the loading amount of Cu is 5% and the loading amount of Pt is 0.3%, it is economical and efficient.

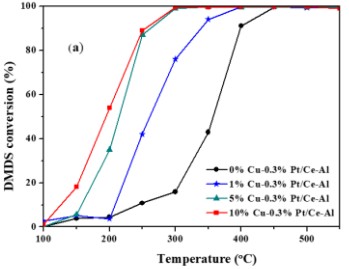 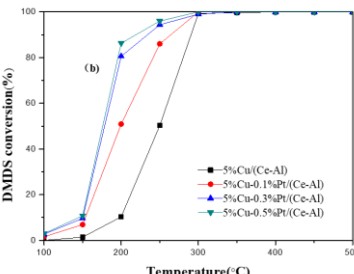

**Figure 11.** (**a**) Activity diagram of catalytic combustion of DMDS by (0%,1%,5%,10%)Cu-0.3%Pt/γ-Al$_2$O$_3$-CeO$_2$, (**b**) Activity diagram of catalytic combustion of DMDS by 5%Cu-(0%,0.1%,0.3%,0.5%)Pt/γ-Al$_2$O$_3$-CeO$_2$.

Since the above results indicate that the Cu-Pt/γ-Al$_2$O$_3$-CeO$_2$ catalyst performance is ideal, we conducted long-term stability experiments, and the stability diagram of catalytic combustion of DMDS for 1000 h is showed in Figure 12. It can be found that the Cu-Pt/γ-Al$_2$O$_3$-CeO$_2$ catalyst maintains a DMDS conversion rate of about 100% and a SO$_2$ yield of 97% at a space velocity of 30,000 h$^{-1}$ for a test period of 1000 h. Cu-Pt/γ-Al$_2$O$_3$-CeO$_2$ catalyst catalyzed combustion of DMDS with good stability and sulfur poisoning resistance is an ideal catalyst for catalytic combustion of DMDS. The possible reason for excellent catalyst stability is the addition of Pt significantly improves the redox ability of Cu-based catalysts, because the adsorption of reactive molecules, the activation of C-H bonds, and the oxidation of SOx during the catalytic combustion of dimethyl disulfide required higher potential. The addition of Pt enabled the Cu-based catalyst to promote the dissociation of O$_2$ at a lower potential, resulting in free O required for SOx oxidation; Dimethyl disulfide was easily partially oxidized during catalytic combustion to form reactive intermediates, such as CO and SOx, which were attached to the active site to poison the Pt-based catalyst as a catalyst [48–52]. However, the addition of Cu can significantly enhanced the desorption and oxidation of CO and SOx by the catalyst, so the catalytic and resistance sulfur poisoning ability of Cu-Pt supported catalyst is excellent by the synergistic effect.

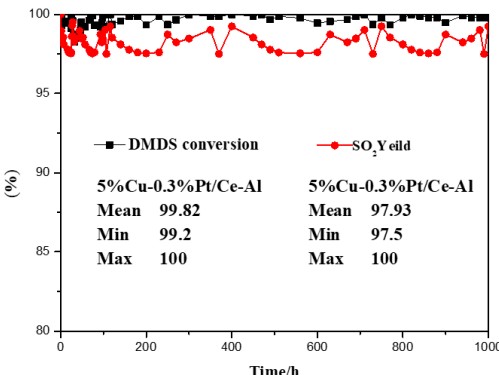

**Figure 12.** Catalytic combustion of DMDS with Cu-Pt/$\gamma$-Al$_2$O$_3$-CeO$_2$ catalyst for 1000 h stability. Experimental condition: airspeed is 50,000 h$^{-1}$, reaction temperature 300 °C, DMDS concentration 1000 ppm, Oxygen concentration 5%.

## 3. Experimental

### 3.1. Catalyst Preparation and Materials

Dimethyl disulfide (CH$_3$SSCH$_3$, AR, Macklin Chemical Company, Tianjin, China). Pseudo-boehmite (AlOOH·nH$_2$O, n = 0.5) and Cerium nitrate hexahydrate (Ce(NO$_3$)$_3$·6H$_2$O, used as a support precursor, was purchased from Zibo Qichuang Chemical Company (Zibo, China). Platinum nitrate, palladium nitrate, copper nitrate, iron nitrate, zinc nitrate, molybdenum nitrate, vanadium nitrate, used as load metal precursor, were purchased from Aladdin Chemical Company (Shanghai, China) and analytical grade.

For the preliminary screening purpose, the catalysts were all prepared by the incipient wetnes impregnation method. Pseudo-boehmite, cerium nitrate or a mixture of them was impregnated with the solution that was prepared by dissolving known amounts of metal salts in deionized water. The mixed solution was magnetically stirred at room temperature for 18 h. The well-impregnated catalyst precursors was spin-dried using a rotary evaporator, dried in air at 120 °C for 24 h, calcined in the oven with the air supply at 550 °C for 4 h, and then ground into particles of 40–60 mesh. The mixture of pseudo-boehmite and cerium nitrate was prepared according to $\gamma$-Al$_2$O$_3$:CeO$_2$ = 10:1 (molar ratio). The metal content in the supported single-metal oxide catalyst is 5 wt%, while each of the supported bimetallic oxides catalyst consists of 5 wt% of Cu and 0.3 wt% of other metals.

### 3.2. Evaluation of the Catalysts

The catalytic activity evaluation process of catalytic combustion DMDS is as follows. First, 0.2 g of catalyst was charged in the middle of the fixed bed reactor (quartz tube reactor with a diameter of 6.0 mm). Then, saturated steam of DMDS was introduced into the main line by nitrogen bubbling, and mixed with nitrogen and air. Finally, combustion reaction takes place in the fixed bed reactor. The reaction conditions are as follows: DMDS concentration is 1000 ppm, GHSV is 50,000 h$^{-1}$, oxygen content is 5%, control temperature is raised from 100 °C to 550 °C, and stable at 5 °C for 20 min. The DMDS and concertration is detected by gas chromatograph (Tokyo, Japan), (Varian CP-3900, FID detector, OV-101 capillary column). The concertration of SO$_2$ is detected by flue gas analyzer (Testo 350, Berlin, Germany).

The conversion of DMDS, and yields of SO$_2$ are defined in the following way

$$\text{DMDS Conversion (\%)} = \frac{C_i - C_0}{C_i} \times 100 \tag{1}$$

$$\text{SO}_2 \text{ Yield (\%)} = \frac{[\text{SO}_2]}{2 x C_i} \times 100 \tag{2}$$

where $C_i$ is the initial feed concentration of DMDS (ppm), $C_0$ is the outlet concentration of DMDS (ppm) and $[SO_2]$ is the concentrations of the corresponding compounds in mol $L^{-1}$.

### 3.3. Catalyst Characterization

The crystal structure analysis of the supported catalyst in this study was characterized by D8FOCUS X-ray diffractometer (XRD, Bruker, Berlin, Germany). The diffractometer (Berlin, Germany) is equipped with Cu K$\alpha$ rays, and the setting parameters are as follows: 2$\theta$ range 5–90°; step 0.1° and dwell time of 1 s. Diffraction patterns were compared to the ICDD database (International Center for Diffraction Data) for the identification of crystalline phases.

Characterization of supported catalyst surface used ESCALAB 250 X-ray photoelectron spectroscopy analyzer (XPS, Thermo Fisher, Washington, USA). The source is Al K$\alpha$, under ultra-high vacuum (UHV), with an X-ray current of 20 mA and a line voltage of 10 kV.

In this study, the oxidation/reduction performance of the catalyst was characterized by a TPR/D/O1100 catalyst fully automatic analyzer (Thermo Fisher Scientific, Washington, USA). The instrument is equipped with a highly sensitive Thermal conductivity detector (TCD) detector (EDAX, Mahwah, NJ, USA). The test procedure is as follows. First, 0.05 g of the catalyst sample is charged into the catalyst fully automatic analyzer (Norcross, GA, USA) for pre-treatment. Pretreatment conditions: maintained at 200 °C for 1 h under a nitrogen atmosphere. Then cooled to room temperature and subjected to $H_2$ temperature programmed reduction. Test conditions: 5% (Vol.) $H_2/N_2$ flow rate 20 mL/min, temperature from 50 °C to 700 °C, heating rate 10 °C /min.

## 4. Conclusions

$\gamma$-$Al_2O_3$-$CeO_2$ was the most optimal support and Cu was observed to be the principal active phase for catalytic combustion of DMDS. Among the six different types of bimetallic supported catalysts, the Cu-Pt/$\gamma$-$Al_2O_3$-$CeO_2$ catalyst exhibits the highest activity and sulfur poisoning ability for the DMDS combustion based on the conversion. According to XRD, $H_2$-TPR and XPS results, there are close interactions between Pt and Cu (intermetallic Pt-Cu) in the nano sized scale, which could be the main reason why this catalyst showed the highest activity.

Under the conditions of GHSV of 50,000 $h^{-1}$, DMDS concentration of 1000 ppm and oxygen concentration of 5%, the prepared 5%Cu-0.3%Pt/$\gamma$-$Al_2O_3$-$CeO_2$ catalyst has the highest catalytic activity of DMDS, and 262 °C can achieve complete conversion of DMDS. In addition, the 5%Cu-0.3%Pt/ $\gamma$-$Al_2O_3$-$CeO_2$ catalyst has a conversion of about 100% in the 1000 h stability test at a space velocity of 30,000 $h^{-1}$, and the $SO_2$ yield is above 97%, which is an ideal catalyst for catalytic combustion of DMDS.

**Author Contributions:** Formal analysis and experiment, J.G.; writing—original draft preparation, S.G.; Investigation, J.W.; writing—review and editing, H.Z.; funding acquisition and experimental design, J.Z.

**Funding:** This research was funded by Beijing science and technology projects: Z181100005418011, National Natural Science Foundation of China: No. 21676021.

**Conflicts of Interest:** The authors declare no conflict of interest.

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
