# Peer review of "Catalytic Combustion of Dimethyl Disulfide on Bimetallic Supported Catalysts Prepared by the Wet-Impregnation Method"

_catalysts, doi:10.3390/catal9120994_

Round 1

Reviewer 1 Report

The manuscript is devoted to bimetallic supported catalysts for catalytic combustion of dimethyl disulfide. This problem is quite relevant, and the choice of objects of study – metals deposited on an oxide matrix – seems reasonable. Work performed at a satisfactory experimental level and described in sufficient detail, which will make it possible, if necessary, to easily reproduce these experiments.

I have a few comments:

P.2, lines 44–48. In these two sentences, the authors contradict themselves. First they write that noble metal supported catalysts are stable to deactivation and recoverable, and then that they have poor resistance to sulfur poisoning. Which statement is true? Authors constantly miss the space before the "degree Celsius" sign. There are also a few more typos associated with spaces in other places (lines 81, 150, 157, 303, 305). In the experimental part, the authors many times mention barium nitrate instead of cerium nitrate.

In general, the article may be of interest to catalytic chemists and, therefore, may be published in Catalysts after a minor revision.

Author Response

Dear Reviewers:

    Thank you for your letter and the reviewers’ comments concerning our manuscript entitled “Catalytic combustion of dimethyl disulfide on bimetallic supported catalysts prepared by the wet-impregnation method”. Those comments are all valuable and very helpful for revising and improving our paper, as well as the important guiding significance to our research. Please see the attachment.

Reviewer 2 Report

The paper by Gao and coworkers presents a screening of different monometallic (and consequently bimetallic) catalysts for catalytic combustion of dimethyl sulfide (DMMS).

While I generally agree on the experimental approach and the data interpretation (see below few detailed comments), I think the overall presentation is weakened by a plenty of typos and grammar mistakes. An improvement in the english writing is strictly required in order to make the paper suitable for publication.

Concerning the scientific contents:

often the authors refer to any curve in a graph as a "spectrum". This is incorrect, spectra are the outcomes of spectroscopy only, whereas here they show XRD and TPD which are not spectroscopic techniques. in all XRD Figures, the minority phases (e.g. CuO, ZnO) are scarcely visible on the graphical point of view, whereas the authors state these are clearly visible. Enlarging the figures (also other ones) will improve their readability. in Figure 2, the XRD patterns show much higher crystallinity/larger particles for two samples (Cu/Ce-Al and Cu-Pt/Ce). Is this correct or a labelling error? Still, the very sharp features are ascribable to the CeO2 phase. This has much larger crystals in these samples, thus it should have much lower specific surface area. How can this impact on the catalytic outcomes? The authors should discuss this point. page 4, line 104-105, the authors state "The complete conversion temperature of Cu/γ-Al2O3-CeO2 is about 308oC" referring to Figure 3, but this temperature refer to 90% conversion (thus not complete). page 4, lines 120-123, the authors ascribes the change in the intensity of XRD patterns among different metals to a reduction/increase of the regularity of the mesopores. This is in my opinion an odd interpretation: apart the technical difficulties in comparing absolute intensities in XRD on different samples, this behavior is observed only when the diffraction lines refer to diffraction from regular voids, e.g. hexagonal packed mesopores in MCM and SBA silicas (see as an example Gatti et al., Phys Chem Chem Phys, 2017, 19, 14114). The supports presented in this work should not present regular mesoporosity according to the synthetic conditions, the changes in the intensities are most probably barely instrumental. page 9, line 225-226, the sentence "The possible reasons are as follows: the specificity of the specific surface area of the catalyst carrier itself is fixed." is completely non sense.   in Section 3.1, barium nitrate is often mentioned. Did the authors mean cerium nitrate hexahydrate? Otherwise the role of barium in the supports synthesis has to be explained.

Author Response

Dear Reviewers,

         Thank you for your letter and the reviewers' comments concerning our manuscript. Please see the attachment,
